# Regeneration of Panniculus Carnosus Muscle in Fetal Mice Is Characterized by the Presence of Actin Cables

**DOI:** 10.3390/biomedicines11123350

**Published:** 2023-12-18

**Authors:** Mariko Hamada, Kento Takaya, Qi Wang, Marika Otaki, Yuka Imbe, Yukari Nakajima, Shigeki Sakai, Keisuke Okabe, Noriko Aramaki-Hattori, Kazuo Kishi

**Affiliations:** 1Department of Plastic and Reconstructive Surgery, Tachikawa Hospital, 4-2-22 Nishiki-cho, Tachikawa-shi, Tokyo 190-8531, Japan; mhamada.2011d@gmail.com; 2Department of Plastic and Reconstructive Surgery, School of Medicine, Keio University, 35 Shinanomachi, Shinjuku-ku, Tokyo 160-8582, Japan; kento-takaya312@keio.jp (K.T.); otakimarika@gmail.com (M.O.); ucarina.0610@gmail.com (Y.N.); shigekix724@hotmail.com (S.S.); dawndawn@hotmail.co.jp (K.O.); nonken@2001.jukuin.keio.ac.jp (N.A.-H.); 3Faculty of Pharmacy, Keio University, 1-5-30 Shiba Koen, Minato-ku, Tokyo 105-8512, Japan; qiw990217@keio.jp (Q.W.); yuka_9109@keio.jp (Y.I.)

**Keywords:** fetal mice, panniculus carnosus muscle, regeneration, scar formation

## Abstract

Mammalian skin, including human and mouse skin, does not regenerate completely after injury; it is repaired, leaving a scar. However, it is known that skin wounds up to a certain stage of embryonic development can regenerate. The mechanism behind the transition from regeneration to scar formation is not fully understood. Panniculus carnosus muscle (PCM) is present beneath the dermal fat layer and is a very important tissue for wound contraction. In rodents, PCM is present throughout the body. In humans, on the other hand, it disappears and becomes a shallow fascia on the trunk. Fetal cutaneous wounds, including PCM made until embryonic day 13 (E13), regenerate completely, but not beyond E14. We visualized the previously uncharacterized development of PCM in the fetus and investigated the temporal and spatial changes in PCM at different developmental stages, ranging from full regeneration to non-regeneration. Furthermore, we report that E13 epidermal closure occurs through actin cables, which are bundles of actomyosin formed at wound margins. The wound healing process of PCM suggests that actin cables may also be associated with PCM. Our findings reveal that PCM regenerates through a similar mechanism.

## 1. Introduction

The damaged skin of aqueous and certain tailed amphibians, such as newts, can completely regenerate, even after they undergo complete metamorphosis [1,2]. Conversely, in adult mammals, injured skin does not regenerate; instead, it heals by forming scar tissue [3]. However, even in mammals, until a certain developmental stage, the skin completely regenerates without leaving a scar after sustaining a wound [4,5,6,7,8]. In wound healing in mice, skin structures, including the panniculus carnosus muscle (PCM), are fully regenerated up to embryonic day 13. However, they are only partially regenerated in fetuses and adults after embryonic day 14 [9,10,11]. The PCM in rodents is distributed in layers across the whole body and is characterized by its cutaneous mobility. The PCM is a skeletal muscle located under the dermal fat layer and above the subcutaneous adipose tissue [12,13]. It is mainly composed of type II fibers. In rodents, PCM is associated with the stimulation of wound contraction [14]. In humans, this muscle remains as facial expression muscles and plays a role in facial expression. However, in the trunk, the PCM disappears and remains as fascial structures and trace tissues, such as the superficial fascia. [15,16]. The subcutaneous adipofascial tissue in humans is divided into two layers by the superficial fascia, with the shallow layer consisting of protective adipofascial tissue called PAFS (protective adipofascial system) and the deeper layer consisting of lubricant adipofascial tissue called LAFS (lubricant adipofascial system) [11,15]. The PAFS acts to protect the shallow part against external forces that reach deeper into the body. The LAFS acts as a lubricant to ensure smooth musculoskeletal movement against the skin. The superficial fascia lies between these two structures.

Partially torn skeletal muscles also regenerate [17,18,19,20]. During this process, muscle satellite cells, which are stem cells in skeletal muscle, are activated [21], proliferate, express MyoD (a master regulator of muscle differentiation), differentiate, and promote regeneration [11,22,23]. MyoD and Myf5 are expressed during the differentiation of myogenic cells into myoblasts. Furthermore, the MyoD-induced expression of transcription factors, such as myogenin and MEF2, leads to the differentiation of myoblasts into myotubular cells [21,24,25,26,27]. In adult mice, experimentally damaged PCM can partially regenerate; however, this regenerative process stops at the site of the open wound, without any fusion occurring between the opposing ends of the muscle. This cessation of PCM regeneration may also be related to the formation of skin scars.

The boundary between PCM regeneration and non-regeneration coincides with the period of skin regeneration. Scar-forming cells in adult mice develop from the fascia beneath the PCM [28,29,30,31]. It is not yet known when the PCM is formed during embryonic life and why it regenerates after damage at E13. We considered that the PCM plays an important role in wound healing and focused on the association between the PCM and wound healing. To gain a clearer understanding of the regenerative processes in the PCM, it is crucial to rigorously compare the pre- and post-phases of the transition from regeneration to scar formation. For this reason, it is also important to understand the developmental stages and developmental processes of the PCM.

Skin regeneration is associated with actin cable formation at the epithelial margin [32]. In various regenerating animal species, actin filament cables connect epithelial cells at the tip of the wound epithelium, extending across and forming a ring around the wound [32,33,34,35]. These actin cables play a crucial role in fetal tissue wound healing, contracting like a purse string. We have previously demonstrated that actin cables arise in E13 wounds and disappear after E15 wounds in the epidermis [9]. We hypothesized that this mechanism contributes to PCM regeneration. Thus, we investigated whether actin cables formed in the PCM after E13 and E15 and their role in PCM regeneration, assuming a similar mechanism is also occurring in the PCM.

AMP-activated protein kinase (AMPK) is also involved in the regulation of cell migration by influencing microtubules and actin filaments. We previously reported that AMPK activation promotes wound regeneration through the formation of actin cables [9]. In this study, we investigated whether AMPK activity contributes to PCM regeneration via actin cable formation. To this end, we evaluated the morphological changes in PCM regeneration induced by acadesine, an AMPK activator, in experimentally induced skin wounds at E14 and E15.

The purpose of this study was to explore the relationship between the PCM and the transition from a regenerative to a non-regenerative phenotype in fetal mouse wound healing. Furthermore, the role of actin cables in the PCM of wounds was assessed before and after this transitional period in mice.

## 2. Materials and Methods

This study protocol was reviewed and approved by the Keio University Institutional Animal Care and Use Committee at the Keio University School of Medicine (approval number: 14090–[4]). All experiments were performed in accordance with the Institutional Guidelines for Animal Experimentation of Keio University.

### 2.1. Mice

All mice were obtained from Sankyo Laboratory Services (Shizuoka, Japan). In this experiment, adult mice and 8-week-old male ICR mice were used (*n* = 8), as well as E11-E15 fetal mice from euthanized pregnant mice (*n* = 8 fetuses per time point). All mice were housed in an environmentally controlled facility maintained at 24 °C and with a 12-h light/dark cycle (7 a.m.–7 p.m.), with water ad libitum.

### 2.2. Wound Model

Wounding was performed at E13, E14, and E15. Pregnant mice were anesthetized with 3% isoflurane, and the skin and abdominal wall was incised by scissors to expose the uterus. Tissue samples were obtained from the entire skin layer of the lateral thoracic region of fetal mice. The procedure for injury induction is described below. All intra-abdominal surgical manipulations were performed using an operating microscope. After the abdominal wall incision, the uterus was carefully withdrawn from the abdominal cavity using non-toothed forceps. The myometrium, amniotic sac, and yolk sac were incised using microsurgical scissors and micro forceps (No. 5, 110 mm). A 2 mm full-layer incision was made in the lateral thoracic region of the fetuses using microsurgical scissors. Microsurgical scissors were used, which are jagged on one side to prevent skin tissue from escaping during microsurgical procedures. Subsequently, the amnion and yolk sac were sutured with 9-0 nylon. On E13 and E14, the myometrium remained open without sutures, and the fetus was returned to the abdominal cavity with the amnion and yolk sac covered. On E15, the fetus was returned to the uterus, and the myometrium was sutured with 9-0 nylon. On E13 and E14, the fetus was allowed to grow within the abdominal cavity. Although suture closure of the myometrium was still possible on E13 and E14, low post-operative fetal survival rate was a problem. Therefore, the method was modified to not suture the uterus on E13 and E14. We confirmed that there was no difference in the state of wound healing of the fetus with and without uterine sutures. During the operation, the fetus, placenta, and abdominal cavity were kept constantly moist by a small amount of PBS to prevent them from drying out. Immediately before closing the wound, mice were intraperitoneally injected with the uterine relaxant ritodrine hydrochloride (Fujifilm Wako Pure Chemical, Osaka, Japan) at a dose of 1 μg/g per body weight. The peritoneum and skin were sutured using 5-0 nylon. Wounds were created in at least four fetuses. The wound was labeled with 0.25% 1,1′-dioctadecy-3,3,3′,3′-tetramethylindocarbocyanine perchlorate (DiI) dissolved in 1% ethanol in phosphate-buffered saline (PBS) to demarcate the wound area upon retrieval of the fetuses. Fetal wounds were harvested at 12, 18, 24, 48, and 72 h after wounding (*n* = 8 fetuses per time point).

To create an adult-wound-healing model, adult mice were administered general 3% isoflurane anesthesia via inhalation. A full-layer incision of approximately 1 cm in the longitudinal axis was made in the lateral thoracic region involving the PCM. They were subsequently euthanized on days 3, 5, 7, and 10 via cervical dislocation (*n* = 8 per time point). The skin, including the wound, was harvested and treated as described below.

Acadesine (Sigma-Aldrich, St. Louis, MO, USA), the AMPK activator, was used to evaluate changes in PCM regeneration by regulating AMPK activity in vivo. E14 and E15 fetuses were subjected to wound surgery and received 100 µL of acadesine (20 mM) in the amniotic fluid. To assess the changes in wound healing, fetuses were harvested 72 h after surgery (*n* = 8 fetuses per time point).

### 2.3. Preparation of Tissue Sections

Tissue samples were harvested from the skin and subcutaneous tissue, including the PCM in the lateral thoracic region. It is difficult to harvest only particular parts of the skin and subcutaneous tissue, especially at an immature stage of development such as E11–E14. Up to E11–E14, tissue samples were harvested, including fascia and ribs. From E15, ribs and skeletal muscle were removed. Tissue samples were collected and fixed overnight in 4% paraformaldehyde. After the fixation of tissue samples, the tissues were processed differently for sectioning:

Paraffin sections (7 µm thick): Fixed tissues were embedded in paraffin and sliced into 7 µm-thick sections.

Frozen Sections (10 µm thick): Fixed tissues were immersed in 20% sucrose/phosphate-buffered saline (PBS) and 40% sucrose/PBS and embedded in OCT compound (Sakura Finetek Japan Co., Ltd., Tokyo, Japan). After embedded, the tissue was frozen and sliced into 10 μm-thick sections.

Whole mount: The tissue samples were prepared with the PCM wound facing toward the cover glass and the epidermal wound facing toward the glass slide.

### 2.4. Staining Tissue Sections

Paraffin sections were soaked in xylene, rehydrated with gradually lower concentrations of ethanol, and transferred to PBS. The sections were incubated in 1×TE buffer (10 mM Tris Base, 1 mM EDTA, pH 9.0) for 60 min at 98 °C.

The tissue sections were stained with hematoxylin and eosin (H&E) (Muto Pure Chemicals Co., Ltd., Tokyo, Japan). To visualize the collagen-rich extracellular matrix, Masson’s trichrome staining was performed (Muto Pure Chemicals Co., Ltd., Tokyo, Japan).

For immunofluorescence staining, the paraffin and frozen sections were blocked with 2% bovine serum albumin (BSA) in PBS for 1 h at room temperature (20~25 °C). The primary antibody used for immunofluorescence staining was an anti-desmin antibody (MA122150; 1:1000; Thermo Fisher Scientific, Waltham, MA, USA). The secondary antibodies utilized included Alexa Fluor 555 (A32727; 1:500; Thermo Fisher Scientific, Waltham, MA, USA) or Cy5 (A10524; 1:500; Thermo Fisher Scientific, Waltham, MA, USA). Before using anti-desmin antibody, the sections were blocked with M.O.M (MKB-2213, Vector laboratories, Newark, CA, USA) for 1 h at room temperature because these primary antibodies were mouse-derived monoclonal antibodies. Nuclear staining was performed using DAPI (D-9542; 1:500; Sigma-Aldrich, St. Louis, MO, USA).

PCNA was visualized using the DAB Substrate Kit (SK-4100; Vector Laboratories, Newark, CA, USA), according to the manufacturer’s instructions. The primary antibody used for immunostaining was an anti-PCNA antibody (sc-56; 1:200; Santa Cruz Biotechnology, Dallas, TX, USA). The secondary antibodies used were anti-mouse IgG antibodies (BA2000, Vector Laboratories, Newark, CA, USA) and the VECTASTAIN^®^ ABC KIT was used (Vector Laboratories, Newark, CA, USA). Color development was performed using a DAB (3,3′-Diaminobenzidine) solution.

Alexa Fluor 488 conjugated phalloidin (A12379; 1:200; Thermo Fisher Scientific, Waltham, MA, USA) was used for staining the polymerized actin. Phalloidin was used in the frozen sections and whole mounts.

### 2.5. Imaging

The stained tissues were photographed using an all-in-one fluorescence microscope (BZ-X700; Keyence, Osaka, Japan) to observe PCM development and the process of regeneration. To visualize and observe the wound and its surroundings, a consolidated, wide-field image was obtained. To observe actin cables, whole-mount tissues were observed under a confocal laser scanning microscope (FV3000; Olympus/FV10-ASW 3.0 Viewer; Olympus, Tokyo, Japan) and analyzed using the FV10-ASW 3.0 Viewer (Olympus, Tokyo, Japan). Morphological images of the wound area were obtained using ImageJ 1.52k software (National Institutes of Health, Bethesda, MD, USA). The wound area was evaluated by measuring the area around the edges of the hole using polygonal sections after importing the photographic images into ImageJ. Actin cable formation was also assessed by investigating the directionality of the actin cables to the wound.

### 2.6. Statistical Analysis

Data in the text are presented as the mean ± SE. Statistically significant differences between the two groups were determined. An unpaired Student’s *t*-test was used to analyze differences between the two groups. Differences were considered significant at *p* < 0.05. A one-way analysis of variance (ANOVA) and post hoc tests (Tukey’s method) were used to compare differences between three or more groups. IBM SPSS Statistics 29 (IBM, Chicago, IL, USA) was used. A *p*-value < 0.05 was considered significant.

## 3. Results

### 3.1. Location of PCM on E13

The PCM, located just below the dermis (Figure 1; surrounded by a dotted line), forms part of the subcutaneous layer of the skin in the trunk, along with loose fascia and deep muscle. To investigate the development of the PCM beneath the dermis, we collected normal skin (epidermis, dermis, and subcutaneous tissue) from fetal mice ranging from E11 to E15. We tracked the developmental progress of the PCM. On E11–E12, we observed the epidermal–dermal structure; however, the PCM layer was not observable under the dermis based on desmin staining (Figure 2a–c). By E12, the skeletal muscles of the trunk developed deep under the skin (Figure 2d–f). On E13, the PCM was identified as a single layer just beneath the dermis (Figure 2g–i, surrounded by a white dotted line). Furthermore, the PCM persisted beyond E14 (Figure 2j–o, delineated by a white or black dotted line).

### 3.2. PCM Regenerates at E13, but Not after E14

Tissue sections of the skin wound from fetuses removed at 12, 18, 24, 48, and 72 h after injury were prepared to observe changes in the PCM at the wound edge (*n* = 8 fetuses per time point) (Figure 3A). Within the first 24 h after injury, no PCM regeneration was observed in the wound at any stage from E13 to E15 (Figure 3A(a–f)). However, 48 h after injury, a substantial difference in PCM regeneration between E13 and E14 was observed. In E13 fetuses 48 h after injury, PCM regeneration led to the merging of severed PCM with the opposite edge, resulting in a continuous PCM without deformities (Figure 3A(g)). In contrast, in E14 and E15 fetuses 48 h post-injury, the PCM did not display continuous regeneration and resembled the PCM state observed at 24 h after injury (Figure 3A(h,i)).

To assess the PCM 48 h after wounding, its ratio of regeneration was assessed (Figure 3B). The length of PCM regeneration was assessed as a ratio to the length of the wound margin. E13 fetuses had a 100% regeneration rate. E15 fetuses clearly had a reduced PCM regeneration rate compared to E13 fetuses (*p* < 0.01). Percent PCM regeneration was calculated according to the following formula: [distance covered by regenrated PCM]/[distance between wound margin] × 100.

Using PCNA, a marker of cell division, we investigated whether the de-differentiation and re-differentiation of muscle satellite cells, which typically occurs during the regeneration of muscle fibers, takes place at the PCM margins after injury in E13 and E15 fetuses (Figure 4). In adult mice, approximately 5–7 days after injury, increased cell division was observed at the wound margins of the PCM (Figure 4B). No increase in cell division confined to the limbus was observed at the wound edge of the PCM in the E13 and E15 fetuses (Figure 4A).

### 3.3. PCM Closure via Actin Cables

On E13, in the epidermis, when the skin had completely regenerated, actin cables formed at the edges of the epidermal tissue. These actin cables exerted tension, leading to contraction of the epidermis, ensuring that skin regeneration maintained the original epidermis–dermis relationship [9]. We used confocal laser microscopy to examine whether actin cables were present at the incised edges of the PCM and epidermis, using fluorescent Phalloidin as a specific marker for F-actin. On E13, 24 h after injury, Phalloidin staining indicated the presence of actin cables at the PCM edge, which are distinct from the actin bundles normally found in muscle fibers (Figure 5A). These actin cables resembled a drawstring bag in E13 wounds at 24 h after injury. In contrast, no actin cables were observed on E15 (Figure 5A(d–i)). On E14, almost no actin cables were observed, although remnants of actin cables seemed to remain. Therefore, wound closure was suggested to align with the orientation of actin cables in the width direction. By quantifying the differences in actin cables between E13 and E15, we assessed the directionality of Phalloidin-stained actin towards the wound. Directionality was assessed ‘up to 45° from the wound surface’ and ‘45~90° from the wound surface’ (Figure 5B). E13 wounds exhibited a higher degree of orientation ‘up to 45° from the wound surface’ and little orientation ‘45~90° from the wound surface’ compared to those at E14 and E15, (*p* < 0.01). Compared to E13 wounds, the directionality of E15 wounds predominantly showed orientation “45~90° from the wound surface” (*p* < 0.01). The striped fluorescence defects in E13 images reflect the shadows of the ribs. In the preparation of the wound samples, the tissues were harvested in blocks from the epidermis to the chest wall. Normally, the ribs are micro-surgically removed from the sample for better imaging, but up to E14, the PCM and ribs were too fragile to separate, leaving their shadow images.

AMPK activation promotes wound regeneration by forming actin cables [9]. In the present study, to investigate whether actin cable formation in the PCM is associated with AMPK activity, we evaluated the effect of AMPK activity on wound healing in the fetal PCM. We observed morphological changes in injured PCM at E14 and E15 using acadesine. Acadesine was administered via intrauterine injection following PCM injury, and the fetuses were removed 72 h post-injury (Figure 6A). To assess the PCM and actin at the wound margin, we observed the tissue containing the wound margin in a whole mount. At E14 72 h post-injury in the AMPK-activated group, PCM closure was incomplete following acadesine treatment (Figure 6B). Similar to the untreated wounds at E13 24 h post-injury (Figure 5A(a–i)), actin cables appeared as contracted drawstring bags with some parts showing mixed actin cable degradation. At 72 h post-injury on E15, the wounds in the AMPK-activated group still displayed PCM defects, albeit with reduced defect sizes (Figure 6B,D). At E15, the wounds in the AMPK-activated group tended to be closer than those in the control group (Figure 6B). Although PCM closure remained incomplete, the effect of acadesine was also observed at E14 (Figure 6C).

## 4. Discussion

In rodents such as mice, the PCM is present throughout the body, whereas in humans, except for the facial muscles, the PCM has disappeared from the trunk, leaving only the superficial fascia as fascial tissue [16]. Skeletal muscles have the ability to regenerate from partial tears, even in adults. This regeneration occurs primarily through the de-differentiation, migration, and re-differentiation of muscle satellite cells, which are the stem cells of skeletal muscle. However, completely severed skeletal muscle lacks a scaffold and does not regenerate. Previously, we reported complete scar-free skin regeneration at E13, contrasting with the absence of such regeneration after E14 [4,9,10,11]. We also observed differences in the occurrence of dermal fibrosis between E15 and E17 [4,9,11,28]. This makes our model ideal for investigating organ morphogenesis mechanisms by comparing the wound healing process at different developmental stages in mice.

In this study, we found a significant difference in the regenerative potential of the PCM, which was incised simultaneously with the skin, between E13 and E14. Although the PCM is located in subcutaneous tissue, its developmental stages have not been investigated in detail. Therefore, in this study, we first analyzed the developmental origin of the PCM at each developmental stage in fetal mice from E11 to E15. At E11, the skin and subcutaneous tissue were very thin, and no clear muscle tissue was observed in the trunk, whereas at E12, skeletal muscle development was observed in the trunk. At E13, we observed the development of the PCM characterized by a single layer of muscle tissue beneath the dermis.

In adult mice, regeneration of the damaged PCM at the wound edge was minimal. However, the direction of regeneration extended through the incised edges of the dermis and granulation tissue toward the epidermis, rather than toward the opposite incised edge. Conversely, in E13 mice, the incised edges of the PCM advanced toward the opposite incised edges, resulting in complete PCM regeneration. Furthermore, when examined over time, PCM regeneration in E13 mice occurred within 48 h after injury, aligning with epidermis regeneration. After E14, regeneration of the PCM ceased prematurely.

We previously reported that the difference in skin regeneration between E13, E14, and later stages lies in the faster closure of the E13 dermis wound compared to the epidermis. Consequently, E13 skin fully regenerates due to undisturbed epidermal–dermal interaction [9]. The regeneration process of PCM coincides with the timing of skin regeneration. Alternatively, it is possible that the deeper layers of the PCM briefly form a dermal scar after E14, filling the dermal defect and inhibiting PCM regeneration, as suggested by previous reports from Takaya et al. [4].

We also considered that regeneration of the incised PCM edges in E13 mice involved actin cables similar to those observed in the epidermis and that contraction of these actin cables leads to complete regeneration of the PCM. When acadesine was administered, actin cables formed at the PCM wound edges, which was followed by a tendency towards PCM closure in both E14 and E15 wounds.

Understanding the mechanism underlying PCM regeneration and identifying ways to restore severed PCM may have implications for treating facial muscle tears caused by blunt trauma in humans. Additionally, it might offer non-surgical avenues for treating congenital anomalies related to abnormal facial muscle development, such as cleft lip.

## 5. Conclusions

The developmental processes of the PCM in fetal mice were visualized. PCM was found to originate from E13. Immunostaining of E13 wounds showed the presence of desmin- and phalloidin-positive actin cables in the PCM and the epidermis. The damaged PCM was observed to regenerate within 48 h after injury in E13 wounds, whereas it was observed that the PCM had not fully regenerated after E15. AMPK activation in E15 wounds improved PCM contraction and decreased the gap after wounding. The role of actin cables in PCM regeneration was suggested.

## Figures and Tables

**Figure 1 biomedicines-11-03350-f001:**
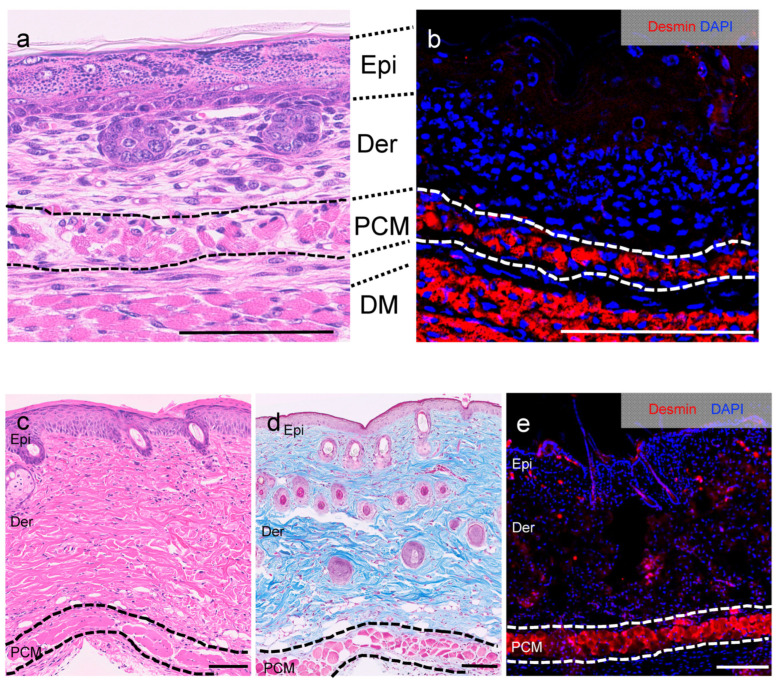
Representative structure of skin and subcutaneous tissue in mice. Tissue samples were obtained from the lateral thoracic region. (**a**) Hematoxylin–eosin (HE) staining of skin and subcutaneous tissue on embryonic day 17 (E17) (*n* = 8). The panniculus carnosus muscle (PCM) is located just below the dermis, as enclosed by the dotted line. (**b**) Immunofluorescence staining of skin and subcutaneous tissue on E17. Muscle tissue was stained using an anti-desmin antibody. The PCM is observable just below the dermis. Red: desmin, blue: DAPI. (**c**) HE staining of skin and subcutaneous tissue in 8-week-old mice (*n* = 8). Black dotted line: PCM. (**d**) Masson’s trichrome skin staining of skin and subcutaneous tissue in adult mice. Black dotted line: PCM. (**e**) Immunofluorescence staining of skin and subcutaneous tissue in adult mice. Muscle tissue is stained using an anti-desmin antibody. Red: desmin, blue: DAPI. White dotted line: PCM. Ep: epidermis; Der: dermis; DM: deep muscle; PCM: panniculus carnosus muscle. Scale bars: 100 μm (**a**–**e**).

**Figure 2 biomedicines-11-03350-f002:**
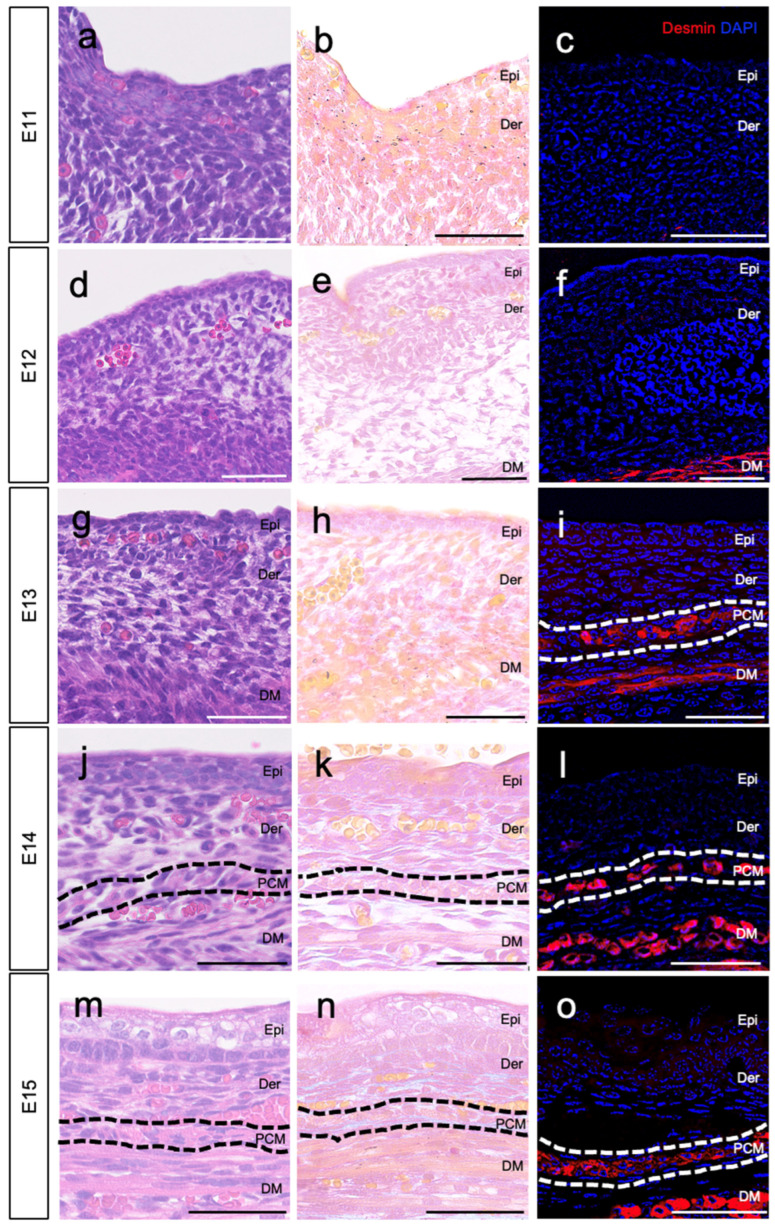
The developmental progress of representative PCM in fetal mice. The development of PCM from E11 to E15 was observed. (**a**–**c**) Muscle tissue could not be observed in the skin and subcutaneous tissue at E11. (**d**–**f**) DM could be observed in the subcutaneous tissue at E12. (**g**–**i**) We observed the development of PCM underneath the dermis at E13. (**j**–**l**) Even after E14, PCM was observed underneath the dermis. (**m**–**o**) On E15, PCM was observed using HE and Masson’s trichrome staining. HE staining (**a**,**d**,**g**,**j**,**m**), Masson’s trichrome staining (**b**,**e**,**h**,**k**,**n**), and immunostaining (**c**,**f**,**i**,**l**,**o**); E11 (**a**–**c**), E12 (**d**–**f**), E13 (**g**–**i**), E14 (**j**–**l**), and E15 (**m**–**o**). Black and white dotted lines indicate PCM. *n* = 8 fetuses per time point. Scale bars: 50 μm (**a**–**o**). Red: desmin, blue: DAPI. DM: deep muscle; PCM: panniculus carnosus muscle.

**Figure 3 biomedicines-11-03350-f003:**
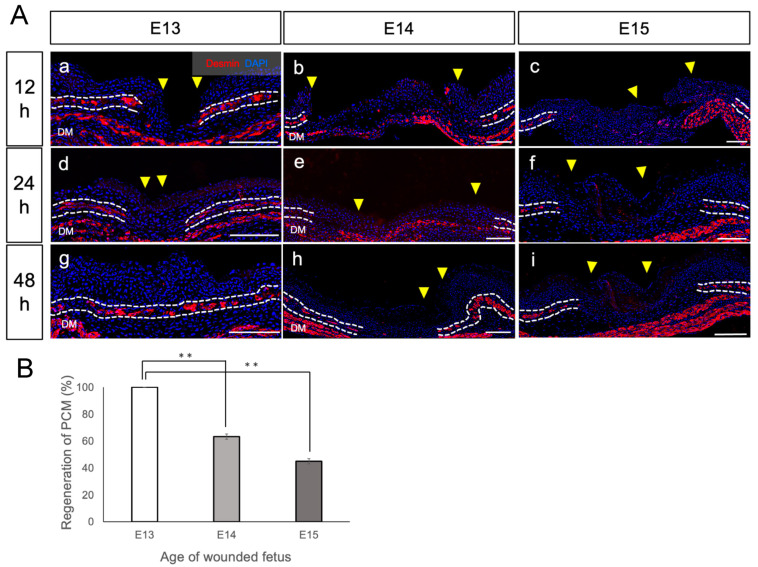
Regeneration process of damaged PCM during the fetal period. Time progression of experimentally damaged PCM at embryonic day 13 (E13), E14, and E15. (**A**) A 2-mm-long whole skin incision wound was performed on the lateral thorax of the fetal mice in their mother’s womb, and PCM regeneration was assessed both temporally and spatially for each embryonic day. The PCM and epidermal wound edges were observed. At E13, the wound edges did not regenerate within the first 24 h after injury, but they did fully regenerate within 48 h after injury. PCM regeneration was not observed after E14. Red: desmin, blue: DAPI. Yellow arrowheads: edges of the damaged epidermis. White dotted line: PCM. *n* = 8 fetuses per time point. Scale bar: 100 μm (**a**–**i**). (**B**) To assess PCM regeneration, the rate of PCM regeneration was quantified. E14 and E15 fetuses clearly had a reduced PCM regeneration rate compared to E13 fetuses (*p* < 0.01). **: *p* < 0.01.

**Figure 4 biomedicines-11-03350-f004:**
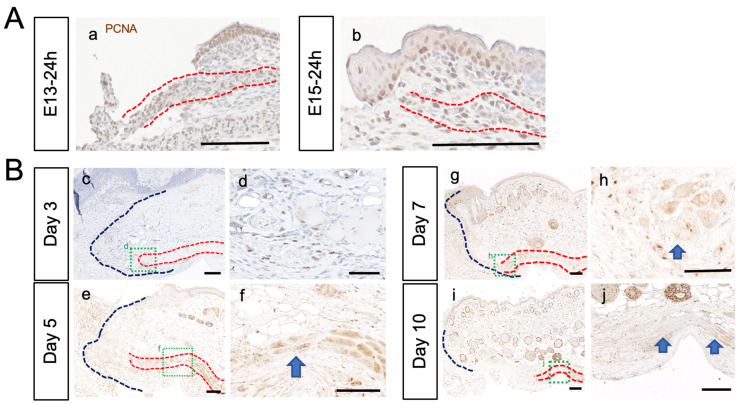
Skeletal muscle de-differentiation and re-differentiation at the PCM wound edges. (**A**) PCM was experimentally injured in fetal mice at E13 and E15, and was observed 24 h post-injury using DAB staining by anti-PCNA antibody. PCM showed overall division at E13 (**a**) and E15 (**b**), with no specific cell division at the margins. Red lines: PCM. Scale bars: 100 μm. (**B**) Experimentally injured PCM in 8-week-old mice 3 (**c**,**d**), 5 (**e**,**f**), 7 (**g**,**h**), and 10 (**i**,**j**) days after injury. Approximately 5–7 days post-injury, the accumulation of mononuclear cells and mitotic figures was observed at the PCM wound margin. Arrow: the accumulation of mononuclear cells and increased mitosis. Red lines: PCM. Scale bars: 100 μm (**c**,**e**,**f**,**g**,**i**,**j**) and 50 μm (**d**,**h**).

**Figure 5 biomedicines-11-03350-f005:**
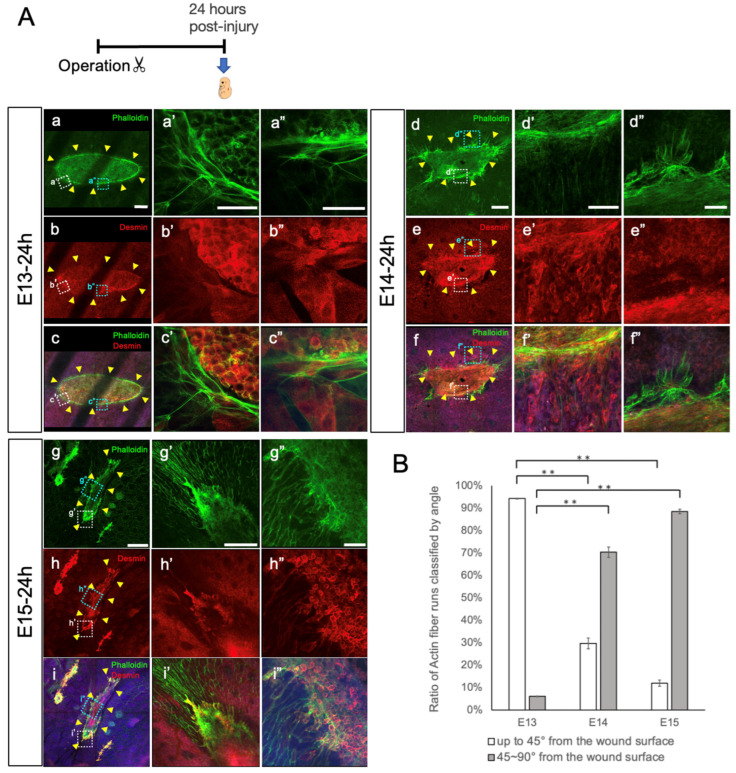
PCM closure via actin cables. (**A**) PCM was experimentally injured in fetal mice at E13, E14, and E15, and was observed 24 h post-injury. Whole-mount sections of skin and subcutaneous tissue showing the PCM 24 h post-injury on E13 (**a**–**c**,**a’**–**c’**,**a”**–**c”**). Whole-mount sections of skin and subcutaneous tissue showing the PCM 24 h post-injury on E14 (**d**–**f**,**d’**–**f’**,**d”**–**f”**).Whole-mount sections of skin and subcutaneous tissue showing the PCM 24 h post-injury on E15 (**g**–**i**,**g’**–**i’**,**g”**–**i”**). At E13, actin cables exhibited a long running structure. The striped fluorescence defects observed in the E13 images correspond to the shadows cast by the ribs. At E14 and E15, actin cables do not exhibit drawstring-bag-like morphology, as at E13. *n* = 8 fetuses per time point. Green: phalloidin, red: desmin, blue: DAPI. Scale bars: 500 µm (**g**–**i**); 200μm (**a**–**f**,**g’**–**i’**); 50μm (**a’**–**f’**,**a”**–**i”**). Yellow arrowheads: wound edges. (**B**) To assess differences in actin cables at E13, E14, and E15, the directionality of actin relative towards the wound was analyzed. The directionality was evaluated by the angle from the wound surface. E13 wounds exhibited a higher degree of orientation ‘up to 45° from the wound surface’ and little orientation ‘45~90° from the wound surface’ compared to E14 and E15 wounds (*p* < 0.01). **: *p* < 0.01.

**Figure 6 biomedicines-11-03350-f006:**
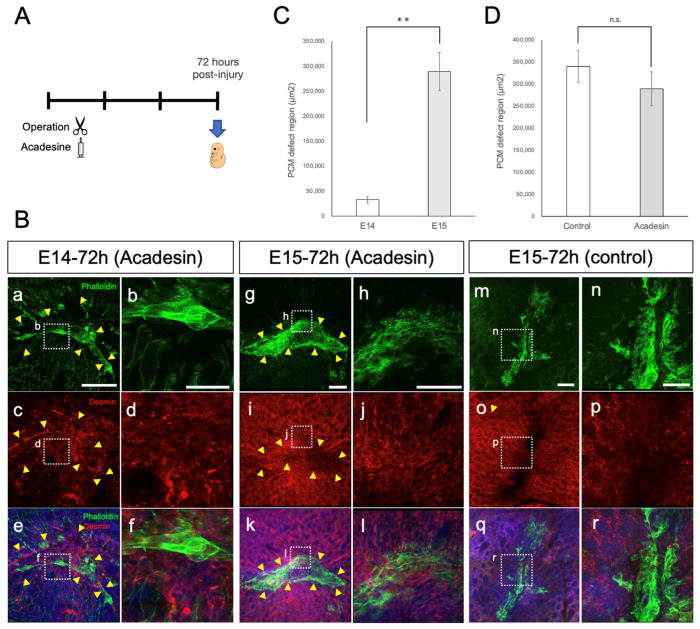
The size of the PCM defect was reduced after the administration of acadesine (the AMPK-activated group). (**A**) Experimental design. (**B**) Wounds of AMPK-activated groups 72 h post-injury on E14 and E15. (**a**–**f**) PCM injury at E14 tended to have a smaller PCM defect size after acadesine treatment. Actin cables formed at the PCM margin. (**g**–**l**) AMPK-activated groups 72 h post-injury on E15 also exhibited PCM contraction. (**m**–**r**) Non-treatment groups 72 h post-injury on E15. Yellow arrowheads: wound edges. *n* = 8 fetuses per time point. Green: phalloidin, red: desmin, blue: DAPI. Scale bars: 200 µm (**a**,**c**,**e**,**g**,**i**,**k**,**m**,**o**,**q**); 50 µm (**b**,**d**,**f**); 100 µm (**h**,**j**,**l**,**n**,**p**,**r**). (**C**) PCM defect region in AMPK-activated groups 72 h post-injury on E14 and E15 (*p* < 0.01). **: *p* < 0.01. (**D**) PCM defect region in AMPK-activated groups and non-treatment groups (control) 72 h post-injury on E15. n.s.: not significant.

## Data Availability

The generated data are contained within the article.

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
