# Peer review of "Regeneration of Panniculus Carnosus Muscle in Fetal Mice Is Characterized by the Presence of Actin Cables"

_biomedicines, 2023, doi:10.3390/biomedicines11123350_

Round 1

Reviewer 1 Report

Comments and Suggestions for Authors

I think that the topic is of interest, but some aspects are really missing. What ist the role of the different fibroblast types. There a major differences between fetal and adult fibroblasts on one side, on the other side there different fibroblasts within the different layers of skin and also on the fascia level. All these aspects are not really mentioned and discussed. A lot of relevant recent papers are not cited and discussed. 

Author Response

Dear Editors and Reviewers

Thank you very much for reviewing our manuscript and offering valuable advice. We are thankful for the time and energy you expended. We have addressed your comments with point-by-point responses and revised the manuscript accordingly. Thank you for giving us the time to reply.

Our responses to the referees’ comments are as follow:

Reviewer1

I think that the topic is of interest, but some aspects are really missing. What is the role of the different fibroblast types. There a major differences between fetal and adult fibroblasts on one side, on the other side there different fibroblasts within the different layers of skin and also on the fascia level. All these aspects are not really mentioned and discussed. A lot of relevant recent papers are not cited and discussed.

Reply:

Thank you very much for your excellent suggestion. The properties of fibroblasts in fetuses differ from those of adult animals and, as noted, their properties are known to vary by site.

Specifically, fibroblasts in the papillary layer of the dermis have been reported to have properties that mainly promote tissue regeneration, whereas fibroblasts in the dermal reticular layer promote scar formation (Driskell & Watt Nature 2013). Recent reports using single-cell RNA sequencing have also shown that fibroblasts in skin can be clustered into more subtypes, representing a population of cells with diverse properties.

However, current our study does not discuss fibroblast diversity, but rather examines the development of Panniculus carnosus muscle layer and its ability to regenerate after injury, which is a different topic from fibroblast diversity and is not discussed in detail.

Reviewer 2 Report

Comments and Suggestions for Authors

The authors represented an interesting point of scar formation. The authors used a skin injury model and studied the skin changes in different developmental stage in fetal mice. While the results are solid, some limitations of this manuscript need to be addressed before acceptance. Some comments are listing below. 

1. The manuscript was not well-prepared. The Title, the abstract, conclusion and the material and method section need to be revised. 

2. The are many qualification results of the tissue staining. Please add quantification data to the relevant data, such as in figures 1-4. 

3. The authors stated that they intended to "explore the relationship between PCM and the transition from a regenerative to a non-regenerative phenotype in fetal mouse wound healing". However, the authors only showed the progress of the PCM generation. No comparision was performed. 

Comments on the Quality of English Language

Need to be improved by a native speaker.

Author Response

Dear Editors and Reviewers,

Thank you very much for reviewing our manuscript and offering valuable advice. We are thankful for the time and energy you expended. We have addressed your comments with point-by-point responses and revised the manuscript accordingly. Thank you for giving us the time to reply.

Our responses to the referees’ comments are as follow:

The authors represented an interesting point of scar formation. The authors used a skin injury model and studied the skin changes in different developmental stage in fetal mice. While the results are solid, some limitations of this manuscript need to be addressed before acceptance. Some comments are listing below.

Reply:

Thank you very much for your important comments.

  1. The manuscript was not well-prepared. The Title, the abstract, conclusion and the material and method section need to be revised.

Reply:

We agree with you and have incorporated this suggestion throughout our paper. Mainly I have modified the abstract, introduction, material and method section and conclusion. I have added description about panniculus carnosus muscles, what has been reported previously and what is not yet known in introduction. In materials and methods, I added to explain to the detailed surgical procedure and handling instructions. Based on this, the summary and conclusions were revised.

  1. The are many qualification results of the tissue staining. Please add quantification data to the relevant data, such as in figures 1-4.

Reply:

Thank you very much for your invaluable comments. As noted, the data taken needs to be further quantified. To assess the regeneration process of PCM, the rate of PCM regeneration was quantified (Figure3). This enabled the impression to be made that there is a significant difference in wound healing of the injured fetus in E13 and E15.

  1. The authors stated that they intended to "explore the relationship between PCM and the transition from a regenerative to a non-regenerative phenotype in fetal mouse wound healing". However, the authors only showed the progress of the PCM generation. No comparision was performed.

Reply:

To assess the observation of wound healing, first, it was shown whether PCM was already present in E13 and the subsequent development of the PCM. Subsequently, the change from a regenerative to a non-regenerative phenotype was observed by creating wounds containing PCM. After showing whether PCM was already present at E13 and the subsequent course of development, we then looked at the change from a regenerative to a non-regenerative phenotype by creating wounds that included PCM. Previous studies have shown that the skin regenerates completely up to E13, but after E14 it is repaired leaving a scar and does not regenerate. The boundary between PCM regeneration and non-regeneration coincides with the period of skin regeneration, leading to the hypothesis that PCM regeneration is associated with skin regeneration and the conception of this study. Takaya et al. reported that skin regeneration involves contraction of actin cables by actin cable attachment, and this paper aims to show that a similar mechanism is involved in PCM regeneration. This study showed the presence of desmin- and phalloidin-positive actin cables in the PCM as well as in the epidermis. Moreover, although phosphorylation of AMPK by acadesin treatment did not cause PCM regeneration in E14 and E15, but it did promote PCM contraction and reduce the extent of the PCM defect.

The corrections have been made in red.

This paper was proof-checked at Editage.

We wish to express our strong appreciation to the reviewers for their insightful comments on our paper. We feel the comments have helped us significantly improve the paper.

Reviewer 3 Report

Comments and Suggestions for Authors

The study by Hamada et al. describes the process of muscle tissue regeneration in a fetal mice injury model. The authors used state-of-the-art methods and presented the results in a brief and concise manner. The manuscript text is well written, and the results and conclusions are easy to follow. However, the manuscript needs a careful revision. The authors are kindly advised to provide more details to some experimental procedures and figures:

2. Materials and Methods

Line 90: Please provide the number of adult animals and fetuses in each study group or experiment.

Line 102: It is important to ensure that the incision depths were not different between E13, E14, and E15? Did the authors expect any variations? Which measures were taken to exclude incision variations in all experiments?

3. Results

Lines 184-194: Please provide the number of analyzed embryos.

Lines 198-207: Please indicate that figures 1a-e and figures 2 a-o are representative for the number of embryos. Are all presented figures obtained from various sections of the same sample?

Line 218: For a comparative statement, it is indispensable that the depth and the gap of the incisions were not significantly different between E13, E14, and E15? On the first sight, the reader would compare the gaps between the edges of the damaged epidermis (yellow arrowheads) and the size of the scale bars. It becomes obvious that the scale bars are much smaller on E14 and E15 than E13. Therefore, the incisions must have been deeper on E14 and E15. Again, please provide the number of embryos for which images 3 a-i are representative.

Line 220: As mentioned in Materials and Methods (lines 103-107), the myometrium was sutured and then returned into the uterus on E15. This is different than on E13 and E15 where the myometrium remained open. The authors should explain why they chose this approach and if this difference can influence the wound closure progress.

Lines 236-243: Figure 4 is not mentioned in the manuscript text.

Lines 259-269: Figure 5 is very important and should be presented at its best. They are several issues that need a more attention. Images 5a, d, and g show diagonal dark strips. If this is caused by technical problems, they should be addressed in the text. Otherwise, the quality of the images should be improved.

Is it possible to label the site of injury by yellow arrowheads or provide HE staining images for a better overview?

Is it possible to provide the images of E14-24h instead of E15-24h? E14-24h would better match the results in figure 6 and avoid the issue with E15 experiments that were treated differently (see my note to line 220).

Diagram 5B should present for E13 and E 15 each 3 bars designating 0~45°, 45~135°, and 135~180°.

Author Response

Dear Editors and Reviewers

Thank you very much for reviewing our manuscript and offering valuable advice. We are thankful for the time and energy you expended. We have addressed your comments with point-by-point responses and revised the manuscript accordingly. Thank you for giving us the time to reply.

Our responses to the referees’ comments are as follow:

Reviewer 3

The study by Hamada et al. describes the process of muscle tissue regeneration in a fetal mice injury model. The authors used state-of-the-art methods and presented the results in a brief and concise manner. The manuscript text is well written, and the results and conclusions are easy to follow. However, the manuscript needs a careful revision. The authors are kindly advised to provide more details to some experimental procedures and figures.

Reply:

Thank you very much for your important comments. We agree with you and have incorporated this suggestion throughout our paper.  I add to explain to the detailed surgical procedure and handling instructions.

  1. Materials and Methods

Line 90: Please provide the number of adult animals and fetuses in each study group or experiment.

Reply:

Added to the text. There are 8 adult mice and 8 fetuses.

Line 102: It is important to ensure that the incision depths were not different between E13, E14, and E15? Did the authors expect any variations? Which measures were taken to exclude incision variations in all experiments?

Reply:

As pointed out, the absence of variation in the depth of injury between E13-E15 fetal ages and the reduction of variation between individuals are considered essential conditions for the accuracy and reliability of the experimental results.

As reported in reference 11, the fat layer between the PCM and the dermis (PAFS; protective adipofascial system) has relatively fine septae and dense connections, whereas the fat layer between the PCM and the deep fascia (LAFS; lubricant adipofascial system) has sparse septal walls, loose connections and a tendency to slide. Therefore, previous experiments have reproducibly shown that the depth of the wound incised by serrated blade scissors reaches the deeper layers of the PCM regardless of the age of the fetus.

  1. Results

Lines 184-194: Please provide the number of analyzed embryos.

Reply:

Added to the text. There are 8 adult mice and 8 fetuses.

Lines 198-207: Please indicate that figures 1a-e and figures 2 a-o are representative for the number of embryos. Are all presented figures obtained from various sections of the same sample?

Reply:

The number of each sample is 8. Figure 1 shows a cross-sectional view of representative fetal and adult skin and subcutaneous tissue after the development of PCM. In Figure 2, a representative developmental process of PCM in a fetus is shown. Each slide was obtained from a separate sample. Added to the text.

Line 218: For a comparative statement, it is indispensable that the depth and the gap of the incisions were not significantly different between E13, E14, and E15? On the first sight, the reader would compare the gaps between the edges of the damaged epidermis (yellow arrowheads) and the size of the scale bars. It becomes obvious that the scale bars are much smaller on E14 and E15 than E13. Therefore, the incisions must have been deeper on E14 and E15. Again, please provide the number of embryos for which images 3 a-I are representative.

Reply:

As for depth, as mentioned above, the incisions at E13~15 all go down to the loose layer of PCM, but the thickness of the skin (epidermis and dermis) gradually thickens during the development phase, so the wound thickness will increase at E14 and 15 compared to E13. There are also differences in the healing process after skin incision at different fetal ages, with the epidermis becoming more fitted and close together as the fetal age progresses. This is a common picture in every sample and is evidence that the depth of the incision is constant. The number of samples is 8 in each case.

Line 220: As mentioned in Materials and Methods (lines 103-107), the myometrium was sutured and then returned into the uterus on E15. This is different than on E13 and E15 where the myometrium remained open. The authors should explain why they chose this approach and if this difference can influence the wound closure progress.

Reply:

To conduct experiment under more physiological conditions, the ideal method for both E13-15 is suture closure of the myometrium after skin incision. Although suture closure of the myometrium is still possible in E13 and E14, the low post-operative fetal survival rate was a problem. Therefore, the method was modified to not suture the uterus in E13 and E14. We have confirmed that there is no difference in the state of wound healing of the survived fetus with and without uterine sutures. This has been added to the methods section.

Lines 236-243: Figure 4 is not mentioned in the manuscript text.

Reply:

Thank you very much for your important suggestion. This has been added to the result section, 3.2.

Lines 259-269: Figure 5 is very important and should be presented at its best. They are several issues that need a more attention. Images 5a, d, and g show diagonal dark strips. If this is caused by technical problems, they should be addressed in the text. Otherwise, the quality of the images should be improved.

Reply:

Thank you very much for your important suggestion. Samples are harvested in block from the epidermis to the chest wall. The sample is placed so that the PCM is in contact with the cover glass and photographed from the PCM side using a confocal microscopy. The black line is the shadow of the ribs. The sample is removed under an operating microscope using a scalpel, surgical micro scissors and micro forceps. Normal samples are removed from the ribs. However, if the fetus is very young, the ribs are very soft and difficult to remove. However, this is only the case when the fetus is young. As the fetus grows larger, the ribs also become larger, and removal of the ribs is more advantageous for PCM imaging. We will add a note about this in the methods section.

Is it possible to label the site of injury by yellow arrowheads or provide HE staining images for a better overview?

Reply:

Thank you for your suggestion. As noted, it was difficult to tell without the markings. Yellow arrowheads are added to the immunostained images to highlight the wound on figure5 and 6.

Is it possible to provide the images of E14-24h instead of E15-24h? E14-24h would better match the results in figure 6 and avoid the issue with E15 experiments that were treated differently (see my note to line 220).

Reply:

Thank you very much for your important comments. E14-24h whole mounts were prepared and imaged with the FV3000 (Figure5). Few actin cables were observed in E14-24h. We have been able to confirm that there is no difference in the healing process when the fetal wound healing occurs while the fetus is in the uterus in E13. The difference in wound healing between E14 and E15 has been confirmed to be unrelated to differences in surgical technique.

Diagram 5B should present for E13 and E 15 each 3 bars designating 0~45°, 45~135°, and 135~180°.

Reply: As indicated, this graph is difficult to understand, so the notation is changed.This assessed the direction of wound contraction in the actinic cable. Since the angle between the direction of the actin cable and the direction of the wound edge is between 0 and 90 degrees, we have changed the classification to two groups, 0 to 45 degrees and 45 to 90 degrees, in the present results.

The corrections have been made in red.

This paper was proof-checked at Editage.

We wish to express our strong appreciation to the reviewers for their insightful comments on our paper. We feel the comments have helped us significantly improve the paper.

Round 2

Reviewer 2 Report

Comments and Suggestions for Authors

The authors addressed most of the comments of the first review report. One more comment is that the quality of the quantified data in the figure are too blur. Please revise. 

Comments on the Quality of English Language

Minor revision is needed. 

Author Response

Dear  Reviewer,

Thank you very much for reviewing our manuscript and offering valuable advice. We are thankful for the time and energy you expended. We have addressed your comments with point-by-point responses and revised the manuscript accordingly. Thank you for giving us the time to reply.

Our responses to the referees’ comments are as follow:

Reviewer 2
The authors addressed most of the comments of the first review report. One more comment is that the quality of the quantified data in the figure are too blur. Please revise. 

Reply:
Thank you for your valuable comments. We have been able to improve the quantification data in the figures. And, the corrections have been made in red.

Reviewer 3 Report

Comments and Suggestions for Authors

I'd like to thank the authors for addressing all my comments in such detail.

Author Response

Dear  Reviewer,

Thank you very much for reviewing our manuscript and offering valuable advice. We are thankful for the time and energy you expended. We wish to express our appreciation to the Reviewer for his or her insightful comments, which have helped us significantly improve the paper. Thank you very much for your important comments.

Mariko Hamada